# Ion permeation pathway within the internal pore of P2X receptor channels

Stephanie W Tam[1], Kate Huffer[1,2], Mufeng Li[1], Kenton J Swartz[1]*

[1]Molecular Physiology and Biophysics Section, Porter Neuroscience Research Center, National Institute of Neurological Disorders and Stroke, National Institutes of Health, Bethesda, United States; [2]Department of Biology, Johns Hopkins University, Baltimore, United States

**Abstract** P2X receptor channels are trimeric ATP-activated ion channels expressed in neuronal and non-neuronal cells that are attractive therapeutic targets for human disorders. Seven subtypes of P2X receptor channels have been identified in mammals that can form both homomeric and heteromeric channels. P2X1–4 and P2X7 receptor channels are cation-selective, whereas P2X5 has been reported to have both cation and anion permeability. P2X receptor channel structures reveal that each subunit is comprised of two transmembrane helices, with both N-and C-termini on the intracellular side of the membrane and a large extracellular domain that contains the ATP binding sites at subunit interfaces. Recent structures of ATP-bound P2X receptors with the activation gate open reveal the unanticipated presence of a cytoplasmic cap over the central ion permeation pathway, leaving lateral fenestrations that may be largely buried within the membrane as potential pathways for ions to permeate the intracellular end of the pore. In the present study, we identify a critical residue within the intracellular lateral fenestrations that is readily accessible to thiol-reactive compounds from both sides of the membrane and where substitutions influence the relative permeability of the channel to cations and anions. Taken together, our results demonstrate that ions can enter or exit the internal pore through lateral fenestrations that play a critical role in determining the ion selectivity of P2X receptor channels.

## Editor's evaluation

This study provides valuable insight into the molecular mechanism of ion selectivity in the broader family of ATP-gated P2X receptors. The experimental data are of high quality, the evidence supporting the conclusions is convincing, and the work will be of broad interest to biophysicists working on ion channel selectivity.

## Introduction

P2X receptors are a family of ion channels that are activated by extracellular ATP (***Khakh and North, 2006***). There are seven subtypes of P2X receptor channels that are widely expressed throughout the body, including the central and peripheral nervous systems, as well as the cardiovascular, immune, respiratory, gastrointestinal, and genitourinary systems (***Khakh and North, 2006***; ***Illes et al., 2021***). They are thought to play a wide range of important roles, from transmitting gustatory signals, sensing bladder filling, mediating neuropathic and inflammatory pain, to regulating immune responses (***Khakh and North, 2006***; ***Surprenant and North, 2009***; ***Schmid and Evans, 2019***; ***Illes et al., 2021***).

Structures of P2X3, P2X4, and P2X7 receptor subtypes solved using X-ray crystallography or cryo-electron microscopy (cryo-EM), reveal that they are trimers with each subunit containing two transmembrane (TM) helices, with both N- and C-termini on the intracellular side of the membrane and a

*For correspondence: swartzk@ninds.nih.gov

**Figure 1.** The cytoplasmic cap in P2X3 receptor channels. (**A**) Side view of the structure of hP2X3$_{Slow}$ with ATP bound (PDB ID: 6ah4). Ribbon representations of each subunit are colored blue, pink, and green, with a HOLE representation along the central axis colored with radii ≤2 Å in brown and radii larger than 2 Å in tan. E11 (corresponding to E17 in rP2X2) is shown in stick representation with carbon (yellow), nitrogen (blue), and oxygen (red). OPM representation of the membrane is shown in light gray spheres and the location of lateral fenestrations at both extracellular and intracellular ends of the pore are indicated with arrows. (**B**) Magnified side view of the transmembrane helices and cytoplasmic cap shown in ribbon representation (left) and with surface representation (right). Residues around the lateral fenestrations are labeled. Asterisk at F7 indicates the side chain was modeled as Ala in the structure. (**C**) Multiple sequence alignment of all human, rat, and mouse P2X subtypes for the region of the cap containing E17 in rP2X2. Residues corresponding to E17 in rP2X2 are outlined with a box, with acidic residues colored red and basic residues colored blue. A full sequence alignment for elements of the cytoplasmic cap is provided in *Figure 2—figure supplement 1* and Uniprot accession numbers are provided in the legend to *Figure 2—figure supplement 2*. (**D**) Intracellular view of the cytoplasmic cap with surface representation. (**E**) View of the cytoplasmic cap from within the pore with surface representation and carbon colored yellow, nitrogen blue, and oxygen red for atoms in E11.

large extracellular domain that contains the ATP binding sites at the subunit interfaces (*Figure 1A*; *Kasuya et al., 2016*; *Kawate et al., 2009*; *Hattori and Gouaux, 2012*; *Mansoor et al., 2016*; *Li et al., 2019*; *McCarthy et al., 2019*). The TM2 helix lines the ion permeation pathway, with the activation gate positioned towards the external end of the pore (*Li et al., 2008*; *Kawate et al., 2009*; *Li et al., 2010*; *Hattori and Gouaux, 2012*). Most P2X receptors are permeable to cations like Na$^+$ and Ca$^{2+}$ (*North, 2002*; *Egan and Khakh, 2004*); however, the P2X5 receptor has been reported to have measurable Cl$^-$ permeability (*Ruppelt et al., 2001*; *Bo et al., 2003*). Although important determinants influencing the relative permeability of Na$^+$ and Ca$^{2+}$ have been identified within the external pore of P2X receptors (*Migita et al., 2001*; *Samways and Egan, 2007*; *Samways et al., 2014*), the regions of the pore determining the relative permeability of cations over anions are unknown.

All the available structures of P2X receptors contain lateral fenestrations between the TM and extracellular domains, providing a pathway for ions to permeate through the external end of the pore (*Kawate et al., 2011*; *Samways et al., 2011*; *Figure 1A*). Although the initial structures of P2X4 receptor channels contained an internal pore where ions could enter or exit at the threefold axis of symmetry (*Kawate et al., 2009*; *Hattori and Gouaux, 2012*), the N- and C-termini in these structures were not resolved. More recent structures of a slowly desensitizing mutant of the P2X3 receptor channel with ATP bound (*Mansoor et al., 2016*) and of the P2X7 receptor channel with and without ATP bound (*Mansoor et al., 2016*) reveal the presence of an intracellular structural element formed by two N-terminal β-strands and a C-terminal α-helix and β-strand that physically caps the cytoplasmic end of the pore (*Figure 1*). Although small lateral fenestrations present in the structures containing these cytoplasmic caps have been proposed to serve as the pathway for water and ions to enter or exit the internal pore (*Mansoor et al., 2016*), these lateral fenestrations would be expected to be largely buried within the membrane given the dimensions of the TM domains in P2X receptor channels (*Mansoor et al., 2016*; *McCarthy et al., 2019*). However, the precise disposition of the lateral fenestrations relative to the membrane remains uncertain because these structures were solved in a detergent solution (*Mansoor et al., 2016*; *McCarthy et al., 2019*).

In the present study, we explored whether the intracellular lateral fenestrations provide a pathway for ions to enter and exit the internal pore. We identified a residue lining the lateral fenestrations that is a conserved acidic residue in the cation-selective P2X1–4 and P2X7 receptor channels but is a conserved basic residue in the anion permeable P2X5 receptor channel and could therefore play a role in determining ion selectivity. When mutated to Cys, this position is accessible to Cys-reactive reagents from both sides of the membrane, and charge-reversing mutations influence the relative permeability of cations to anions. Our results demonstrate that the lateral fenestrations provide an ion permeation pathway at the intracellular end of the pore and that they contain a critical determinant of ion selectivity.

## Results

The objective of the present study was to investigate how ions permeate through the internal pore of P2X receptor channels given the presence of the cytoplasmic caps recently identified in P2X3 and P2X7 receptor channels (*Mansoor et al., 2016*; *McCarthy et al., 2019*). Although the structures of P2X receptors solved thus far are remarkably similar within the large extracellular domain and relatively small TM domain (*Kasuya et al., 2016*; *Kawate et al., 2009*; *Hattori and Gouaux, 2012*; *Mansoor et al., 2016*; *Li et al., 2019*; *McCarthy et al., 2019*), the cytoplasmic cap has only recently been resolved in P2X3 and P2X7 structures (*Mansoor et al., 2016*; *McCarthy et al., 2019*; *Figure 1*; *Figure 2*; *Figure 2—figure supplement 1*). The slowly desensitizing human P2X3 receptor structural construct (hP2X3$_{Slow}$) where the cap was resolved contained three mutations (T13P, S15V, V16I) within the N-terminal region corresponding to those found in the slowly desensitizing rat P2X2 (rP2X2) receptor (*Hausmann et al., 2014*; *Mansoor et al., 2016*), enabling an ATP-bound open state to be resolved, as opposed to a desensitized state that was otherwise captured without the P2X3$_{Slow}$ mutations (*Mansoor et al., 2016*). P2X7 receptors are slowly desensitizing and contain a cytoplasmic ballast unique to that subtype that is positioned below the cytoplasmic cap, likely helping to stabilize the cap (*McCarthy et al., 2019*).

In ATP-bound open state structures of both P2X3$_{Slow}$ and P2X7 receptor channels, the cytoplasmic cap would be expected to prevent ions from permeating at the threefold axis of the channel (*Figure 1D and E*; *Figure 2—figure supplement 1*). While the intracellular lateral fenestrations formed by the TM helices and the cytoplasmic cap are substantial in size, the presence of a lipid membrane would likely diminish the water-accessible surface of the fenestrations considerably (*Figure 1A and B*). We began by using the OPM server (*Lomize et al., 2012*) to predict the boundaries of the lipid membrane, which suggests that the intracellular lateral fenestrations are largely buried within the membrane (*Figure 1B*; membrane boundaries depicted with planes of gray dots). Many of the residues lining the edges of the lateral fenestrations are hydrophobic in P2X receptors (*Figure 1B*, right), consistent with this structural element residing mostly within the lipid membrane.

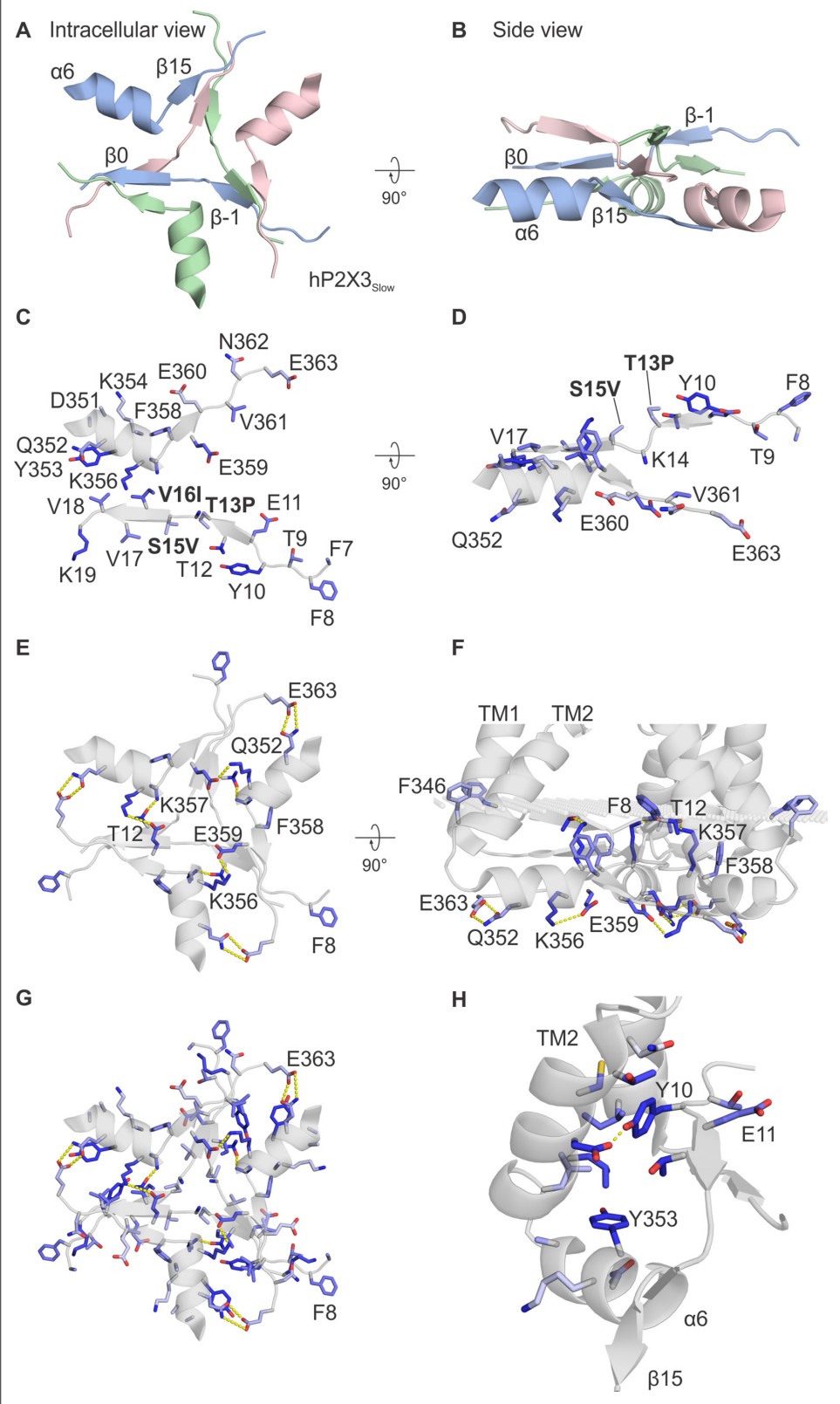

**Figure 2.** Conservation of residues in the cytoplasmic cap in P2X receptor channels. (**A**) Intracellular view and (**B**) side view of the cytoplasmic cap from hP2X3Slow in complex with ATP (PDB ID: 6ah4) using ribbon representations. (**C**) Intracellular view and (**D**) side view showing a single subunit of hP2X3Slow with side chains shown as sticks and colored according to the alignment quality score calculated from the multiple sequence alignments in *Figure 2—*

*Figure 2 continued on next page*

*Figure 2 continued*

**figure supplement 1**, where highly conserved residues are colored in blue and poorly conserved residues are colored in white. Alignment quality score calculated in Jalview based on BLOSUM 62 scores. Residue numbering corresponds to the hP2X3 sequence, and three mutated residues (T13P, S15V, and V16I) that slow desensitization in the construct used for structure determination are highlighted in bold. (**E**) Intracellular view and (**F**) side view of the cytoplasmic cap from hP2X3$_{Slow}$ showing intersubunit side chain interactions between K356 and E359 (3.2–3.8 Å apart); T12 and K357 (2.2–2.3 Å); and Q352 and E363 (2.8–3.8 Å) in yellow. Also pictured are aromatic side chains F8, F346, and F358 facing into the membrane. (**G**) Intracellular view of all three subunits of the cytoplasmic cap of hP2X3$_{Slow}$ with side chains shown as sticks and colored according to the alignment quality score calculated from the multiple sequence alignments in *Figure 2—figure supplement 1*. (**H**) View from the lateral fenestration of hP2X3$_{Slow}$ of two conserved tyrosine residues (Y10 and Y353) and the side chains of residues within 4.0 Å. Y10 and D340 of the neighboring subunit are 2.1–2.2 Å apart. The same orientations were used for panels A, C, E, and G and for panels B, D, and F, respectively.

The online version of this article includes the following figure supplement(s) for figure 2:

**Figure supplement 1.** Conservation of the cytoplasmic cap in P2X receptor channels.

**Figure supplement 2.** Multiple sequence alignment of human, rat, and mouse P2X receptor channels.

## Conservation of the cytoplasmic cap

To explore whether the cytoplasmic cap is a conserved structural element in P2X receptor channels, we constructed a multiple sequence alignment using all seven subtypes and examined the extent to which residues forming the cytoplasmic cap are conserved (*Figure 2*; *Figure 2—figure supplement 1A*; *Figure 2—figure supplement 2*). Overall, residues within the three β-strands (β–1, β0, and β15) and one α-helix (α6) forming the cytoplasmic cap are remarkably well-conserved (*Figure 2C*; *Figure 2—figure supplement 1A*). In the structure of hP2X3$_{Slow}$ with ATP bound, three pairs of residues form hydrogen bonds, including K356 and E359, T12 and K357, and Q352 and E363 (*Figure 2E and F*). In addition, although the side chain of K14 was not well-resolved, it is positioned nearby to E359 and is another candidate for participating in stabilizing hydrogen bonds within the cap. All of these polar residue interactions occur at subunit interfaces, and most of the participating residues are well-conserved across all subtypes of P2X receptors (*Figure 2E and F*; *Figure 2—figure supplement 1A and C*). The cytoplasmic cap also contains two conserved Tyr residues (Y10 and Y353), each of which is surrounded by hydrophobic residues likely to contribute to stabilizing hydrophobic interactions (*Figure 2H*). In the case of Y10, these interactions occur at subunit interfaces, whereas in the case of Y353 the interactions are within each individual subunit, and again these interactions are well-conserved across different P2X receptors (*Figure 2H*; *Figure 2—figure supplement 1A and D*). Y10 also forms a well-conserved intersubunit hydrogen bond interaction with D340, located at the base of the TM2 helix of the neighboring subunit (*Figure 2H*; *Figure 2—figure supplement 1A and D*). We also identified a series of additional aromatic residues (F8, F346, and F358) positioned near to where the OPM server would position the polar headgroup region of the inner leaflet of the membrane, and again these are well-conserved across different subtypes of P2X receptors (*Figure 2F*). All of these stabilizing interactions observed in the cytoplasmic cap of hP2X3$_{Slow}$ are also seen in the structure of hP2X7 (*Figure 2—figure supplement 1A–D*) and the participating residues are conserved in most subtypes of P2X receptors (*Figure 2—figure supplement 1A*), suggesting that structural elements related to the caps seen in P2X3 and P2X7 are likely present in all P2X receptor channels.

## Accessibility of a conserved Glu in the lateral fenestrations of P2X2 receptor channels

In studying the structure of the cytoplasmic cap and intracellular lateral fenestrations in hP2X3$_{Slow}$, we noticed the presence of a Glu residue (E11; *Figure 1*) at the intracellular edge of the lateral fenestration that is conserved in most P2X receptor channels (*Figure 1C*), with the exception of both P2X5 and P2X6, which contain a Lys at this position. Although P2X6 channels do not express as homomeric channels (*Lê et al., 1998*) and, therefore, their permeation properties are poorly understood, homomeric P2X5 receptors can be expressed and have been reported to have a significant Cl$^-$ permeability (*Ruppelt et al., 2001*; *Bo et al., 2003*). We, therefore, thought this position in P2X receptors might provide a foothold to begin interrogating whether the lateral fenestrations contribute to forming an ion permeation pathway and whether residues in this region influence ion selectivity.

To explore whether ions might permeate through the lateral fenestrations, we decided to investigate the accessibility of an introduced Cys residue to thiol-reactive methanethiosulfonate (MTS) compounds. The rP2X2 receptor channel has been extensively used for accessibility studies because it slowly desensitizes and, therefore, is well-suited for examining changes in accessibility between open and closed states (*Jiang et al., 2001*; *Li et al., 2008*; *Li et al., 2010*; *Kawate et al., 2011*; *Heymann et al., 2013*). In addition, the rP2X2–3 T construct, wherein three native Cys residues were substituted with Thr residues, is insensitive to MTS compounds unless additional Cys residues are introduced and have been extensively used for accessibility studies (*Li et al., 2008*; *Li et al., 2010*; *Kawate et al., 2011*; *Heymann et al., 2013*). We began by introducing a Cys into rP2X2–3 T at the position equivalent to E11 in hP2X3 (E17 in the rP2X2), reasoning that if this acidic residue lines the lateral fenestrations in P2X2 receptor channels, a Cys introduced at this position would be accessible to thiol-reactive compounds. The concentration-dependence for activation of the E17C mutation in rP2X2–3 T by free ATP is similar to rP2X2–3 T (*Figure 3—figure supplement 1*), suggesting that the mutant does not dramatically alter the gating mechanism of the receptor. We initially explored the accessibility of 2-trimethylaminoethyl methanethiosulfonate (MTSET) applied from the extracellular side of the membrane to E17C, either when channels are open in the presence of ATP (*Figure 3A and C*), or when they are closed in the absence of ATP (*Figure 3B and D*). Since MTSET has a fixed positive charge at neutral pH, it cannot cross the membrane and only reacts when a Cys is positioned within an aqueous environment within the channel protein (*Holmgren et al., 1996*). Consistent with previous studies (*Li et al., 2008*), when we applied MTSET to the external solution for cells expressing the background rP2X2–3 T construct, we observed no discernible effect of the reagent when applied in the absence or presence of ATP (*Figure 3A, B, E and F*). In contrast, the application of MTSET in the presence of ATP resulted in robust and irreversible inhibition of the E17C rP2X2–3 T channel (*Figure 3C and E*). Application of external MTSET in the absence of ATP had no discernible effect on subsequent current activation by external ATP (*Figure 3D and F*), suggesting that the MTS reagent can only access E17C through the ion permeation pathway from the extracellular side of the membrane when the channel is open. To confirm that MTSET accessibility to E17C requires activation by ATP, we confirmed that the application of MTSET in the presence of ATP produced robust inhibition following the application of the MTSET in the absence of ATP (*Figure 3D and F*). We did not attempt to reverse the inhibitory effects of MTSET using reducing agents as there are five conserved disulfide bonds in the extracellular domains of P2X receptor channels (*Mansoor et al., 2016*; *McCarthy et al., 2019*) that are important for channel trafficking or function (*Clyne et al., 2002*; *Ennion and Evans, 2002*), and in our experience reducing agents make recordings unstable. From these results, we conclude that a Cys residue introduced at E17 within the intracellular lateral fenestrations is accessible to MTSET applied from the external solution when the pore of the channel has been opened with external ATP.

We next studied modification by 2-aminoethyl methanethiosulfonate (MTSEA), which, unlike MTSET, can readily cross membranes (*Holmgren et al., 1996*) and, therefore, provides the means to assess the accessibility of E17C from the intracellular side of the membrane. Similar to MTSET, the external application of MTSEA was without discernible effect on rP2X2–3 T (*Figure 4A, B, E, F*). Also similar to MTSET, external application of MTSEA in the presence of ATP produced robust and irreversible inhibition of ATP-activated current for the E17C rP2X2–3 T channel (*Figure 4C, D and E*), suggesting that external MTSEA can access E17 through the ion permeation pathway. However, in contrast to MTSET, external application of MTSEA in the absence of ATP produced discernible inhibition of currents activated by subsequent application of ATP (*Figure 4D and F*). Inhibition produced by external application of MTSEA in the absence of ATP prevented subsequent inhibition by external application of MTSEA in the presence of ATP (*Figure 4D and F*), suggesting that external MTSEA can access E17C through the ion permeation pathway when the channel is open, or cross the membrane and react with E17C from the intracellular side when the channel is closed.

Having observed robust current inhibition by both MTSET and MTSEA, we next measured the rates of modification of E17C by MTSET and MTSEA in the presence of ATP and compared them with those measured in previous studies where rates of modification were reported for Cys substitutions within the pore-lining TM2 helix (*Li et al., 2008*). The time courses for MTS modification in the presence of ATP could be reasonably well fit by a single exponential function (*Figure 5A and B*), from which we calculate that MTSET modified E17C with a rate of $3.7 \times 10^2$ M$^{-1}$s$^{-1}$, while the smaller MTSEA modified that position with a rate of $2.6 \times 10^3$ M$^{-1}$s$^{-1}$ (*Figure 5D*). From previous MTSET modification

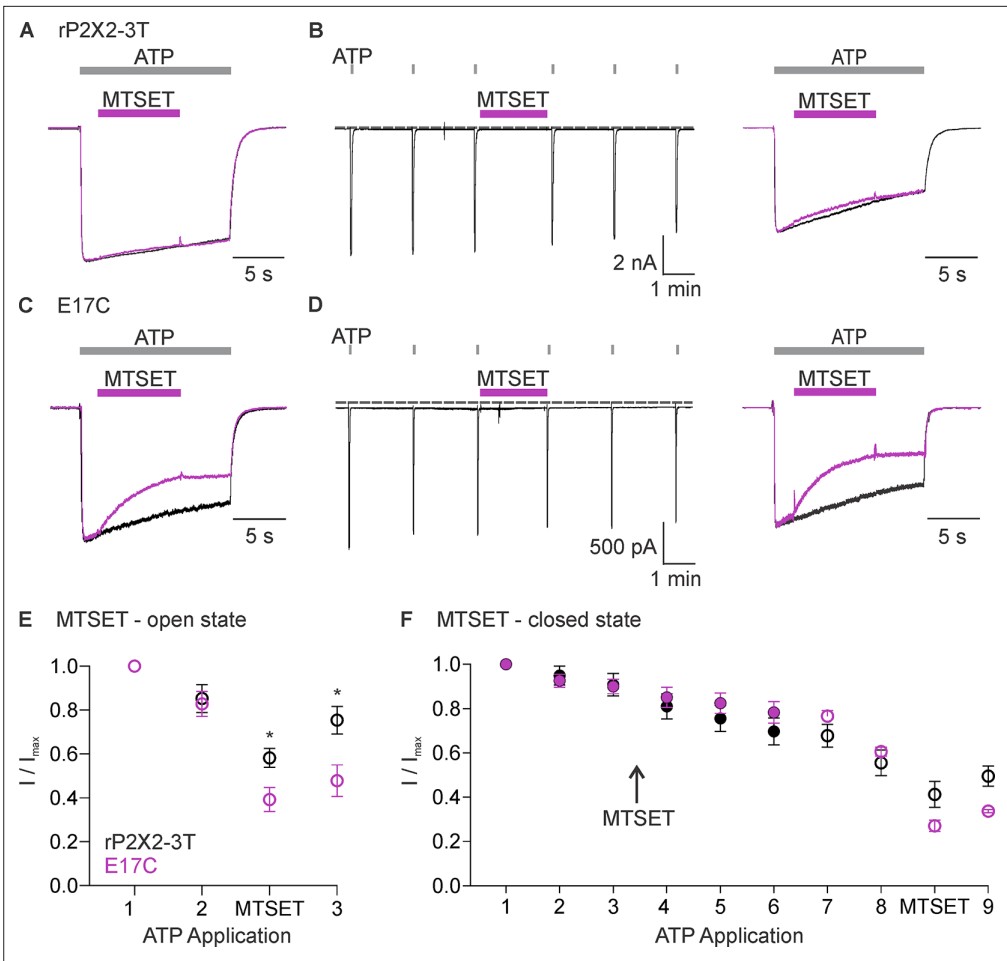

**Figure 3.** Accessibility of E17C in rP2X2–3 T to extracellular 2-trimethylaminoethyl methanethiosulfonate (MTSET). (**A**) Testing for modification of rP2X2–3 T by extracellular MTSET after opening channels with extracellular ATP. Consecutive current traces elicited by extracellular ATP application at -60 mV without (black trace) or with subsequent application of 1 mM extracellular MTSET (purple trace). ATP (1 μM) was applied for 15 s at 2 min intervals three times with MTSET application only during the second ATP application. Superimposed and scaled current traces are shown for the first and second applications of ATP. MTSET (1 mM) inhibited ATP-activated currents by 26 ± 6% (n=4). (**B**) Testing for modification of rP2X2–3 T by extracellular MTSET when channels are closed. ATP (1 μM) was applied six times for 2 s at 2 min intervals and MTSET (1 mM) was applied in the absence of ATP after the third ATP application. Holding voltage was -60 mV. Following the application of MTSET in the closed state, MTSET was applied again in the presence of ATP to the same cell to serve as a control, using the same protocol described in A. MTSET (1 mM) inhibited ATP-activated currents by 14 ± 4% (n=4) when applied in the closed state and by 26 ± 3% (n=3) when subsequently applied in the presence of ATP. (**C**) Testing for modification of E17C rP2X2–3 T by extracellular MTSET after opening channels with extracellular ATP. Consecutive current traces elicited by extracellular ATP application at -60 mV without (black trace) or with subsequent application of 1 mM extracellular MTSET (purple trace). Same protocol as in A. MTSET (1 mM) inhibited ATP-activated currents by 53 ± 5% (n=5). (**D**) Testing for modification of E17C rP2X2–3 T by extracellular MTSET when channels are closed. Same protocol as in B. MTSET (1 mM) inhibited ATP-activated currents by 4 ± 2% (n=7) when applied in the closed state and by 55 ± 5% (n=3) when subsequently applied in the presence of ATP. (**E**) Normalized ATP current amplitudes for MTSET application in the presence of ATP for rP2X2–3 T (black symbols; n=3) and E17C rP2X2–3 T (purple symbols; n=5) using the protocols illustrated in panels A and C. ATP application one corresponds to the peak current amplitude for the first control ATP application within 1 s of applying ATP, two corresponds to the peak current amplitude immediately before MTS application for the second ATP application, measured within 1 s of ATP application. The ATP application labeled as MTSET corresponds to the current amplitude 10 s into the second ATP application at the end of the methanethiosulfonate (MTS) application. Three corresponds to the peak current amplitude during the third ATP application measured within 1 s of applying ATP. (**F**) Normalized ATP current amplitudes during MTSET application in the absence of ATP. Filled black circles (n=3) are from experiments

*Figure 3 continued on next page*

*Figure 3 continued*

with rP2X2–3 T and filled purple circles (n=5) are from experiments with E17C rP2X2–3 T using the protocols illustrated in panels **B** and **D**. MTSET was applied in the absence of ATP between ATP applications three and four as indicated. Open symbols are ATP current amplitudes when testing for open-state modification by MTSET after testing for closed-state modification. Open black circles (n=3) for rP2X2–3 T and open purple circles (n=3) for E17C rP2X2–3 T using the protocols illustrated on the right in panels **B** and **D**. For open state modification following closed state modification, application seven is a control ATP application, and MTSET was applied during application 8 along with ATP. Data point eight is the initial peak current immediately before MTSET was applied and the subsequent data point is at the end of the MTSET application in the presence of ATP after the reagent has had time to react, as illustrated in the right panel of B and D. Data shown in E and F are mean ± SEM, some error bars are smaller than symbols. *p<0.05 by unpaired *t*-test.

The online version of this article includes the following figure supplement(s) for figure 3:

**Figure supplement 1.** Concentration-dependence for activation of rP2X2 constructs.

rates obtained in the presence of ATP for multiple positions within the TM2 helix, we can appreciate that rates of modification diminish as Cys residues are introduced deeper into the pore (*Figure 5D*). Notably, the modification rate for externally applied MTSET at E17C is only incrementally slower than the deepest position in the transmembrane region when applying the reagent to the external solution (*Figure 5D*). Given that the reagents must traverse the entire TM pore to react with E17C, these relatively rapid rates of modification suggest that E17C is positioned in a high dielectric environment that is favorable for modification. Although, lipid molecules likely occupy the upper regions of the lateral fenestrations, it seems unavoidable that the most intracellular end of the lateral fenestrations, where E17 is located, resides in an aqueous environment. From these experiments examining the modification of E17C within the lateral fenestrations, we conclude that this residue lines the intracellular end of the ion permeation pathway.

## Influence of lateral fenestrations on cation selectivity of P2X2 receptor channels

Having uncovered evidence that E17 lines the ion permeation pathway, we next explored whether this conserved acidic residue is a determinant of the cation selectivity in P2X2 receptors. Although, P2X receptor channels are permeable to $Na^+$ and to varying extents $Ca^{2+}$ (*Migita et al., 2001*; *Samways and Egan, 2007*; *Samways et al., 2014*), we are unaware of attempts to measure the relative permeability of cations to anions for most subtypes, except for those that have noted a measurable anion permeability for P2X5 receptor channels (*Ruppelt et al., 2001*; *Bo et al., 2003*). We began by assessing the relative permeability of wild-type rP2X2 to cations over anions by initially measuring equilibrium reversal potentials ($V_{rev}$) using ramps and switching between symmetrical solutions containing 140 mM NaCl on both the external and internal sides and asymmetrical solutions containing 140 mM NaCl internally and 40 mM NaCl externally, using either sucrose or glucose to maintain osmolarity (*Figure 6A*). To minimize errors due to possible ion accumulation on the intracellular side (*Li et al., 2015*), we maintained the cell in symmetrical solutions and only assessed the impact of lowering external NaCl with ATP applications just long enough to measure current-voltage (I-V) relations and determine $V_{rev}$. For wild-type rP2X2 receptor channels, we measured $V_{rev}$ with symmetric solutions near 0 mV and upon switching to the external containing 40 mM NaCl observed that $V_{rev}$ shifted to around –32 mV (*Figure 6A and G*), from which we calculate a $pNa^+$:$pCl^-$ of 20:1 using the Goldman-Hodgkin-Katz equation (see Materials and methods), indicative of a strong preference for cations over anions. Considering that rP2X2 is inwardly rectifying and thus passes outward currents less favorably than inward currents (*Zhou and Hume, 1998*; *Fujiwara and Kubo, 2004*; *George et al., 2019*), we also used voltage step protocols and measured the initial tail current on stepping from –60 mV to more positive voltages, enabling the measurement of relatively linear instantaneous I-V relations and an independent determination of $V_{rev}$ (*Figure 6B and C*). As with the ramp protocol, the step protocol resulted in shifts in $V_{rev}$ from nearly 0 mV to near –32 mV upon switching the external solution from 140 mM NaCl to one containing 40 mM NaCl (*Figure 6B, C and G*). Collectively, these results with either ramp or step protocols establish that rP2X2 exhibits a strong preference for cations over anions.

To explore whether E17 contributes to the cation selectivity, we mutated this residue to Lys, the residue that occupies the equivalent position in anion-permeable P2X5 receptor channels (*Figure 1C*).

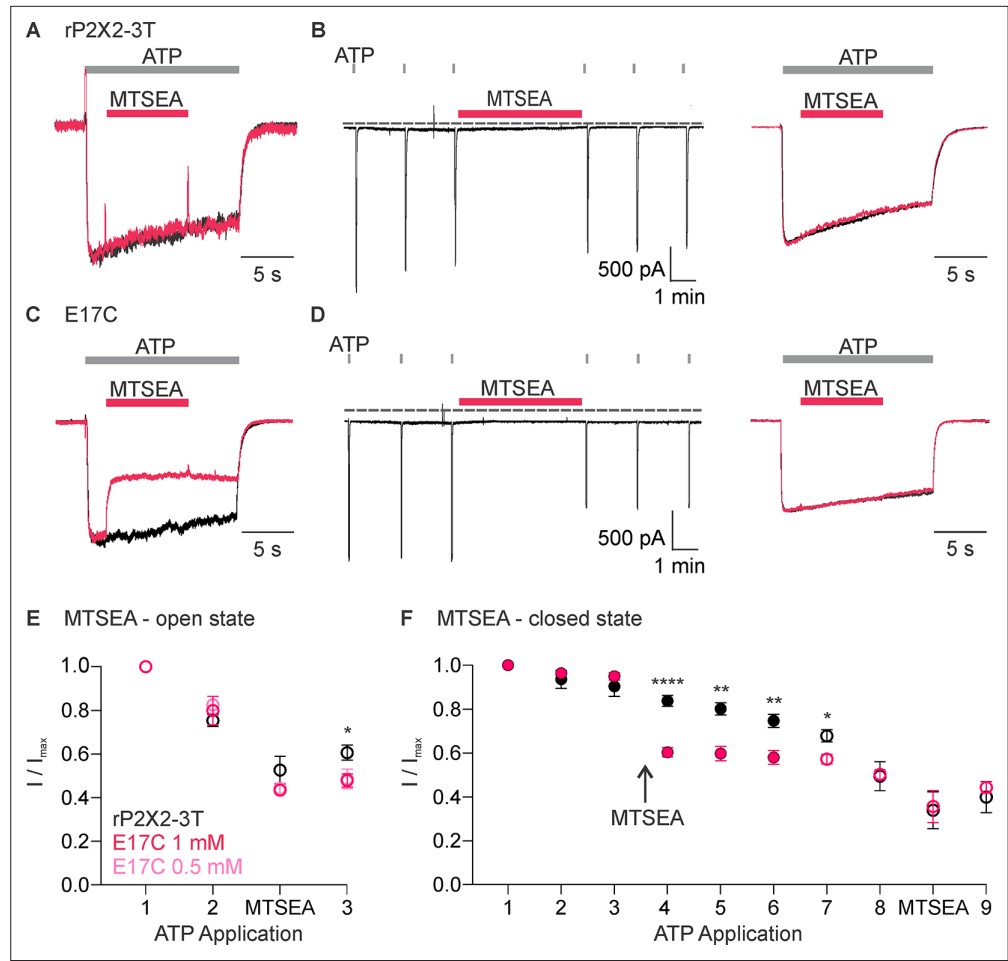

**Figure 4.** Accessibility of E17C in rP2X2–3 T to extracellular 2-aminoethyl methanethiosulfonate (MTSEA). (**A**) Testing for modification of rP2X2–3 T by extracellular MTSEA after opening channels with extracellular ATP. Consecutive current traces elicited by extracellular ATP application at -60 mV without (black trace) or with subsequent application of 1 mM extracellular MTSEA (red trace). ATP (1 μM) was applied for 15 s at 2 min intervals three times with MTSEA application only during the second ATP application. Superimposed and scaled current traces are shown for the first and second applications. MTSEA (1 mM) inhibited ATP-activated currents by 28 ± 6% (n=5). (**B**) Testing for modification of rP2X2–T by extracellular MTSEA when channels are closed. ATP (1 μM) was applied six times for 2 s at 2 min intervals and MTSEA (1 mM) was applied in the absence of ATP after the third ATP application. Holding voltage was -60 mV. Following the application of MTSEA in the closed state, MTSEA was applied again in the presence of ATP to the same cell to serve as a control. MTSEA (1 mM) inhibited ATP-activated currents by 7 ± 2% (n=7) when applied in the closed state and by 34 ± 9% (n=3) when subsequently applied in the presence of ATP. (**C**) Testing for modification of E17C rP2X2–3 T by extracellular MTSEA after opening channels with extracellular ATP. Current traces elicited by extracellular ATP application at -60 mV without (black trace) or with subsequent application of 1 mM extracellular MTSEA (red trace). Same protocol as in A. This protocol was conducted in MTSEA concentrations of both 0.5 mM and 1 mM MTSEA. MTSEA (0.5 mM) inhibited ATP-activated currents by 46 ± 2% (n=4) and at 1 mM inhibited ATP-activated currents by 45 ± 4% (n=4). (**D**) Testing for modification of E17C rP2X2–3 T by extracellular MTSEA when channels are closed. Same protocol as in B. MTSEA (1 mM) inhibited ATP-activated currents by 31 ± 4% (n=6). when applied in the closed state and by 30 ± 13% (n=3) when subsequently applied in the presence of ATP. (**E**) Normalized ATP current amplitudes for MTSEA application in the presence of ATP for rP2X2–3 T (black symbols; n=5) and E17C rP2X2–3 T with 1 mM MTSEA (red symbols; n=4) or 0.5 mM MTSEA (pink symbols; n=4) using the protocols illustrated in panels A and C. ATP application one corresponds to the peak current amplitude for the first control ATP application, two corresponds to the peak current amplitude immediately before methanethiosulfonate (MTS) application for the second ATP application, measured within 1 s of ATP application. The ATP application labeled MTSEA corresponds to the current amplitude 10 s into the ATP application and at the end of the MTSEA application during the second ATP application. Three corresponds to peak current amplitude 2 min after MTS application during the third ATP application, measured

*Figure 4 continued on next page*

*Figure 4 continued*

within 1 s of ATP application. Statistical significance of \*p<0.05 by unpaired *t*-test for 1 mM MTSEA E17C rP2X2–3 T and rP2X2–3 T background construct. (**F**) Normalized ATP current amplitudes during MTSEA application in the absence of ATP. Filled black circles (n=6) are from experiments with rP2X2–3 T and filled red circles (n=4) are from experiments with E17C rP2X2–3 T using the protocols illustrated in panels B and D. MTSEA was applied in the absence of ATP between ATP applications three and four as indicated. Open symbols are ATP current amplitudes when testing for open-state modification by MTSEA after testing for closed state modification. Open black circles (n=3) for rP2X2–3 T and open red circles (n=3) for E17C rP2X2–3 T using the protocols illustrated in panels B and D. For open state modification following closed state modification, application seven is a control ATP application, and MTSEA was applied during application eight along with ATP. Data point eight is the initial peak current before MTSEA was applied and the subsequent data point is at the end of the MTSEA application in the presence of ATP after the reagent has had time to react, as illustrated in the right panel of B and D. Data shown in E and F are mean ± SEM, some error bars are smaller than symbols. \*p<0.05, \*\*p<0.01, and \*\*\*\*p<0.0001 by unpaired *t*-test.

The E17K mutation did not appreciably alter the concentration-dependence for activation of the channel by ATP (*Figure 6H*), suggesting that the mutation does not perturb gating. However, the E17K mutation did alter the shift in $V_{rev}$ produced by lowering external NaCl concentration, shifting $V_{rev}$ to between –22 mV and –25 mV depending on whether we used ramp or step protocols, and whether ions were substituted with sucrose or glucose (*Figure 6D–G*). If we calculate the relative permeability of cations to anions, the E17K diminished $pNa^+$:$pCl^-$ to 10:1 from 20:1 measured for the wild-type receptor. Although these results establish that E17 is not the sole determinant of cation selectivity in P2X2 receptor channels, mutation of this residue detectably alters cation selectivity, consistent with this residue residing within the ion permeation pathway on the intracellular side of the membrane.

## Influence of lateral fenestrations on anion permeability of P2X5 receptor channels

The P2X5 receptor channel has not been extensively studied because heterologous expression results in only small macroscopic currents in response to ATP application (*Bo et al., 2003*; *Sun et al., 2019*; *Illes et al., 2021*). We initially expressed P2X5 receptor channels from rat and mouse but observed that expression of the mouse clone produced the largest ATP-activated currents in our standard extracellular solution lacking divalent cations (see Materials and methods). We also observed a robust

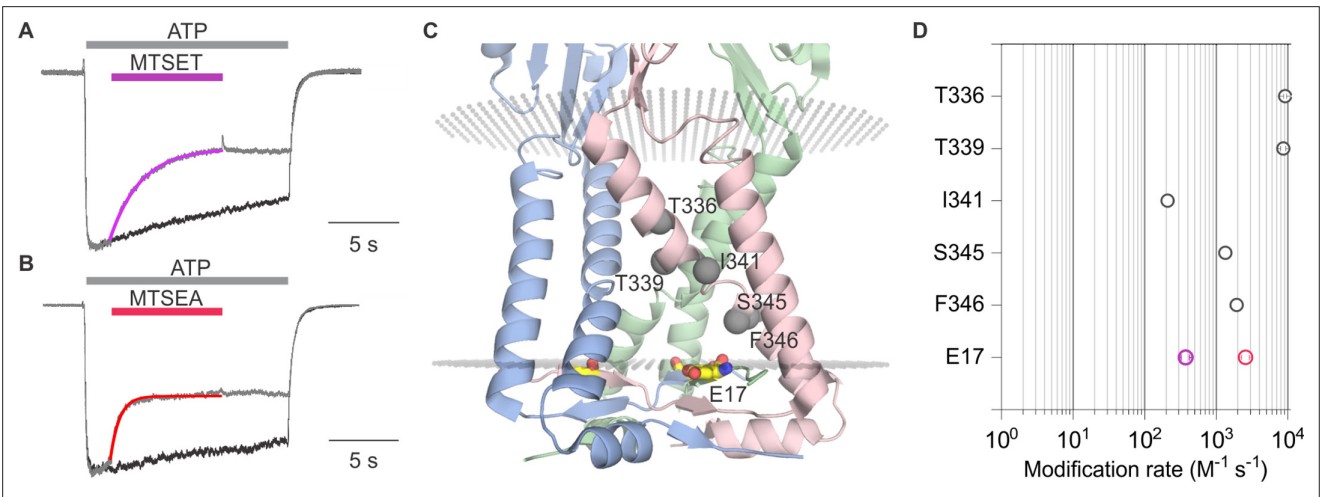

**Figure 5.** Rates of methanethiosulfonate (MTS) modification in rP2X2 for E17C and comparison with rates for the pore-lining TM2 helix. (**A**) Fit of a single exponential function (purple relation) to the time course of current inhibition by 2-trimethylaminoethyl methanethiosulfonate (MTSET) (1 mM) for E17C rP2X2–3 T in the presence of ATP (1 μM) at -60 mV. (**B**) Fit of a single exponential function (red relation) to the time course of current inhibition by 2-aminoethyl methanethiosulfonate (MTSEA) (0.5 mM) for E17C rP2X2–3 T in the presence of ATP (1 μM) at -60 mV. (**C**) Side view of the transmembrane helices and cytoplasmic cap from the structure of hP2X3$_{slow}$ with ATP bound (PDB ID: 6ah4). MTS-accessible residues along TM2 are labeled and each residue alpha carbon is represented as a gray sphere. (**D**) Comparison of MTS modification rates measured in the presence of ATP for E17C rP2X2–3 T (purple open circles for MTSET and red open circles for MTSEA) with MTSET reactive residues in TM2 (gray open circles) from *Li et al., 2008*. Data is mean ± SEM with n=5 for MTSET (1 mM) and n=4 for MTSEA (0.5 mM). Some error bars are smaller than symbols.

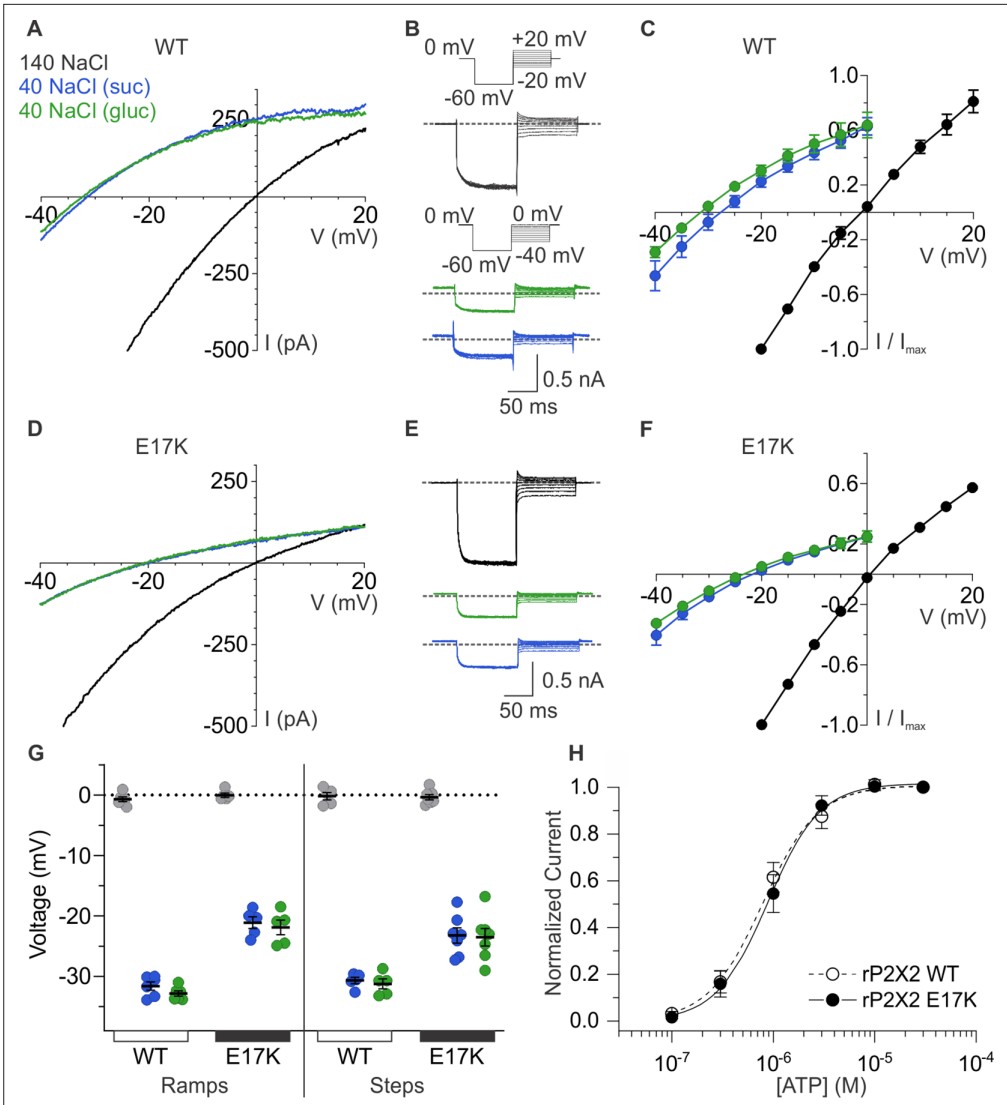

**Figure 6.** Ion selectivity of rP2X2 and influence of the E17K mutation. (**A**) Representative I-V relationships for WT rP2X2 obtained from the same cell using 0.5 s voltage ramps from –60 mV to +60 mV from a holding potential of –60 mV in symmetric 140 mM NaCl solution (black) and two asymmetric NaCl solutions (internal: 140 mM NaCl; external:40 mM NaCl with glucose in green, or 40 mM NaCl with sucrose in blue). Voltage ramps were applied in the absence and presence of 30 μM ATP in each of the three external solutions and currents from ramps recorded in the absence of ATP were subtracted from currents in the presence of ATP to obtain the ATP-activated currents shown. (**B**) Representative current traces from a cell expressing WT rP2X2 elicited using voltage step protocols in symmetrical 140 mM NaCl solution and two asymmetrical solutions as in A. Voltage steps were elicited from a holding potential of 0 mV to –60 mV for 100 ms before stepping to a series of voltages in increments of 5 mV. The range of voltages tested for symmetrical solutions was –20 mV to +20 mV and for asymmetrical solutions was –40 mV to 0 mV. ATP-activated currents were obtained by subtracting control currents recorded in the absence of ATP from currents collected in the presence of 30 μM ATP. The gray dotted line represents 0 pA. (**C**) Normalized instantaneous I-V relationships for WT rP2X2 were obtained using voltage step protocols illustrated in panel B in symmetrical 140 mM NaCl and two asymmetrical solutions. Instantaneous current was measured at 0.5–1 ms following the step from –60 mV to the range of voltages indicated. Data points represent the mean steady-state current ± SEM (n=6 for each condition). (**D**) Representative I-V relationships for the E17K mutant of rP2X2 obtained from the same cell in symmetrical 140 mM NaCl and two asymmetric solutions using the voltage ramp protocol from panel A. (**E**) Representative current traces from a cell expressing the E17K mutant of rP2X2 elicited using the same voltage step protocols from panel B in symmetrical 140 mM NaCl and two external solutions containing 40 mM NaCl. (**F**) Normalized instantaneous I-V relationships for the E17K mutant of rP2X2 were obtained using voltage step protocols illustrated in panel E in symmetrical 140 mM NaCl and two external solutions containing

*Figure 6 continued on next page*

*Figure 6 continued*

40 mM NaCl. Protocols were identical to those in C and data points represent the mean steady-state current ± SEM (n=7). (**G**) Reversal potential measurements for WT and E17K rP2X2 in symmetric and asymmetric NaCl solutions using either ramp or step protocols. Mean and SEM are indicated with bars and individual measurements are indicated with filled-colored circles using the same color code as in A. WT rP2X2 ramp protocol reversal potentials were –0.7 ± 0.4 mV in symmetric solution, –31.6 ± 0.7 mV in asymmetric solution with sucrose, and –32.8 ± 0.4 mV in asymmetric solution with glucose. E17K rP2X2 ramp protocol reversal potentials were 0 ± 0.4 mV in symmetric solution, –21.1 ± 1.0 mV in asymmetric solution with sucrose, and –21.9 ± 1.2 mV in asymmetric solution with glucose. WT rP2X2 step protocol reversal potentials were –0.2 ± 0.6 mV in symmetric solution, –30.7 ± 0.5 mV in asymmetric solution with sucrose, and –31.2 ± 0.8 mV in asymmetric solution with glucose. E17K rP2X2 step protocol reversal potentials were –0.4 ± 0.4 mV in symmetric solution, –23.3 ± 1.3 mV in asymmetric solution with sucrose, and –23.5 ± 1.5 mV in asymmetric solution with glucose. For WT, ramp protocols n=6 and step protocols n=5. For E17K, ramp protocols n=5 and step protocols n=7. (**H**) Normalized concentration-dependence for ATP activation of WT rP2X2 (open circles and dashed line, n=4) and the E17K mutant of rP2X2 (black circles, n=5). Smooth curves are fits of the Hill equation to the data with $EC_{50}$ and $n_H$ values of 0.8 ± 0.1 µM and 1.7 ± 0.1 for WT and 0.9 ± 0.1 µM and 1.6 ± 0.1 for E17K.

yet variable sensitization phenomenon whereby ATP-activated currents in cells expressing mP2X5 increased in amplitude over the time course of several minutes (**Figure 7A and B**). Although we have not yet explored the underlying mechanism, we adopted a standard sensitization protocol (**Figure 7A**) wherein we applied a saturating concentration of ATP (3 µM) until the ATP-activated current reached a steady-state before undertaking further experiments. Examination of the concentration-dependence for activation of mP2X5 revealed a relatively high affinity for ATP in the absence of divalent ions, roughly two orders of magnitude higher than wild-type rP2X2 (**Figure 7C**).

We next examined the relative permeability of mP2X5 for cations and anions using the same approach we employed for rP2X2, in this case only using ramps because we observed relatively linear I-V relations for mP2X5 with little, if any, evidence of inward rectification (**Figure 8A**). Remarkably, decreasing the concentration of external NaCl did not shift $V_{rev}$ towards negative voltages, as we observed for rP2X2, but instead produced a slight positive shift in $V_{rev}$ by a few mV (**Figure 8A and D**). From these shifts in $V_{rev}$, we calculate that mP2X5 has a slight preference for anions over cations, with a relative permeability of $pNa^+:pCl^-$ of 0.8. These findings confirm earlier reports that P2X5 receptor channels have a much higher $Cl^-$ permeability compared to other subtypes of P2X receptor channels (**Ruppelt et al., 2001**; **Bo et al., 2003**).

To explore whether K17 is a determinant of the ion selectivity of mP2X5 receptor channel, we mutated this position to either Glu or Asp. Neither mutation appreciably altered the concentration-dependence

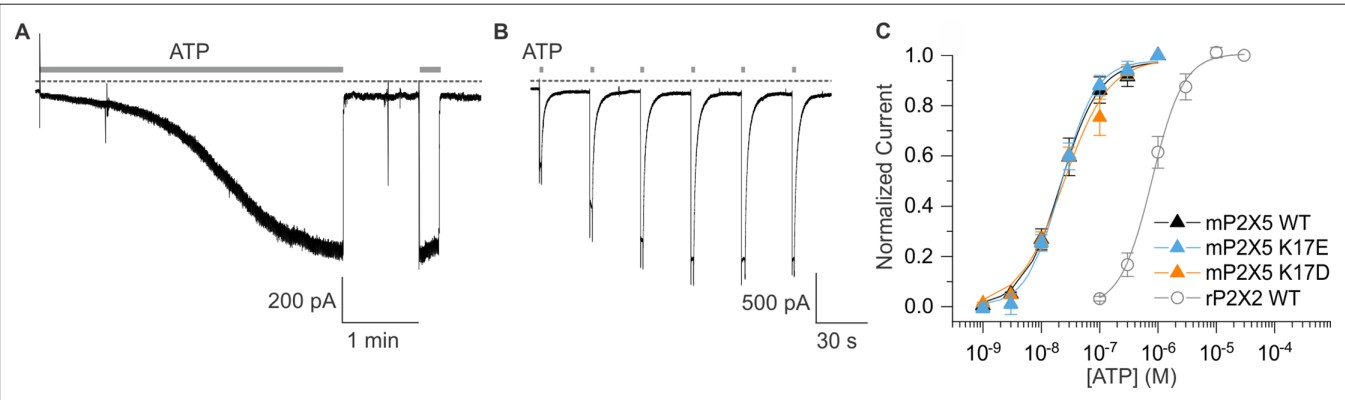

**Figure 7.** Expression and characterization of mP2X5. (**A**) mP2X5 sensitization upon initial exposure to a sustained external application of ATP (3 µM). Subsequent short ATP applications activate slowly desensitizing currents of similar amplitude compared to that achieved during sensitization. Holding voltage was –60 mV. Gray dotted line represents zero current. (**B**) Sensitization was observed while delivering short pulses of saturating ATP (100 µM) given in 30 s intervals. Holding voltage was –60 mV. Gray dotted line represents zero current. (**C**) Concentration-dependence for ATP activation of WT mP2X5 (black filled triangles, n=3), the K17E mutant of mP2X5 (light blue triangles, n=4), and the K17D mutant of mP2X5 (orange triangles, n=3). Smooth curves are fits of the Hill equation to the data with $EC_{50}$ and $n_H$ values of 22.5 ± 1.6 nM and 1.3 ± 0.1 for WT, 21.9 ± 1.7 nM and 1.5 ± 0.1 for K17E, and 24.1 ± 4.3 nM and 1.1 ± 0.2 for K17D. Data for WT rP2X2 are shown for comparison from **Figure 6H**.

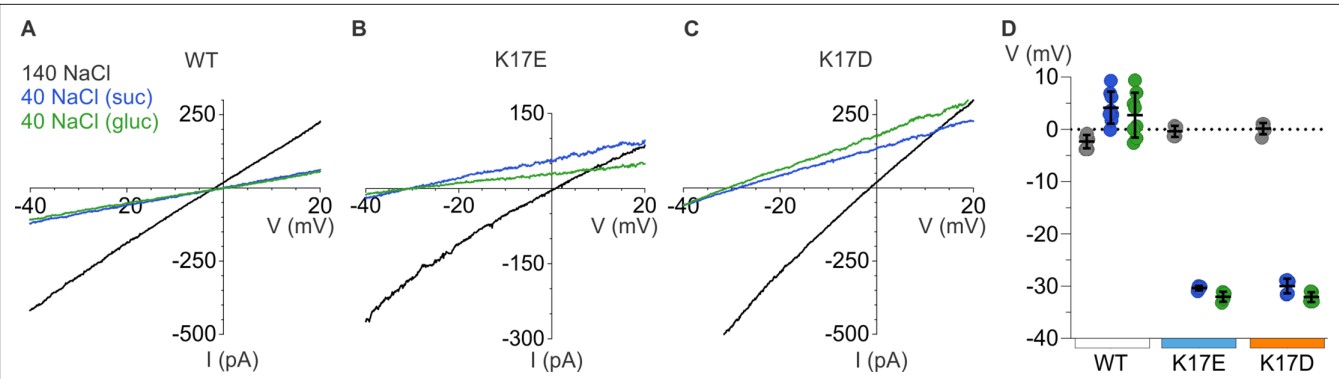

**Figure 8.** Ion selectivity of mP2X5 and influence of K17E and K17D mutations. (**A**) Representative I-V relationships for WT mP2X5 obtained from the same cell using 0.5 s voltage ramps from –60 mV to +60 mV from a holding potential of –60 mV in symmetric 140 mM NaCl solution (black) and two asymmetric NaCl solutions (internal: 140 mM NaCl; external:40 mM NaCl with glucose in green, or 40 mM NaCl with sucrose in blue). Voltage ramps were applied in the absence and presence of 3 μM ATP in each of the three conditions and currents from ramps recorded in the absence of ATP were subtracted from currents in the presence of ATP to obtain the ATP-activated currents shown. (**B**) Representative I-V relationships for the K17E mutant of mP2X5 were obtained from the same cell using conditions as in A. (**C**) Representative I-V relationships for the K17D mutant of mP2X5 were obtained from the same cell using conditions as in A and B. (**D**) Reversal potential measurements for WT, K17E, and K17D mP2X5 in symmetrical and asymmetrical NaCl solutions using ramp protocols. Mean and SEM are indicated with bars and individual measurements are indicated with filled-colored circles using the same color code as in A. WT mP2X5 reversal potentials were –2.3±0.4 mV in symmetric solution, 4.1 ± 1.1 mV in asymmetric solution containing sucrose, and 2.8 ± 1.5 mV in asymmetric solution containing glucose. K17E mP2X5 reversal potentials were –0.4 ± 0.5 mV in symmetric solution, –30.4 ± 0.2 mV in asymmetric solution containing sucrose, and –32 ± 0.5 mV in asymmetric solution containing glucose. K17D mP2X5 reversal potentials were 0.2 ± 0.5 mV in symmetric solution, –30 ± 0.6 mV in asymmetric solution containing sucrose, and –32.1 ± 0.4 mV in asymmetric solution containing glucose. n=8 for WT, n=4 for K17E, and n=5 for K17D.

for activation of mP2X5 by external ATP (**Figure 7C**), suggesting that as for rP2X2, mutations at this position within the lateral fenestrations do not appreciably alter gating. We then proceeded to measure shifts in $V_{rev}$ upon switching external solutions from 140 mM NaCl to 40 mM NaCl, and strikingly observed that for both mutants, lowering the external NaCl concentration produced dramatic shifts of $V_{rev}$ to voltages between –32 mV and –35 mV (**Figure 8B–D**), similar to what we observed for wild-type rP2X2 receptor channels (**Figure 6A–C and G**). From these shifts in $V_{rev}$, we calculate that K17E and K17D mutants are strongly cation-selective, with $pNa^+:pCl^-$ of around 20:1, indistinguishable from what we measured for wild-type rP2X2 receptor channels. These findings indicate that K17 is a critical determinant of the unusual anion permeability of mP2X5 receptor channels. Collectively, our results support the idea that the lateral fenestrations observed in recent structures of P2X3 and P2X7 receptor channels contribute to the intracellular end of the ion permeation pathway.

## Discussion

The objective of the present study was to explore whether ion permeation through the internal pore of P2X receptor channels could occur through the lateral fenestrations seen in recent structures of P2X3 and P2X7 receptors. One enigmatic feature of the lateral fenestrations is that they were observed for detergent-solubilized proteins, and one might expect that they would be partially or completely buried within the lipid bilayer when the protein resides in a membrane environment, in agreement with many residues at the edges of the lateral fenestrations having hydrophobic side chains (**Figure 1**). However, an acidic residue Glu at the intracellular edge of the lateral fenestrations is conserved in P2X receptors thought to be cation-selective but is substituted with the basic residue Lys in P2X5 (**Figure 1**), the one subtype of P2X receptor reported to also be anion permeable. This correlation between side chain character and ion selectivity suggested that this residue might face the ion-conducting pathway and contribute directly to ion selectivity. For the E17C mutant in rP2X2 receptors, we found that the positively charged MTSET could readily react with this position, but only when the channel was opened by external ATP (**Figure 3**). In contrast, the membrane-permeant MTSEA could react with E17C either when the channel was opened by external ATP, or when applied in the closed state in the absence of ATP (**Figure 4**), indicating that E17C is accessible from both

sides of the membrane. Although approximations of P2X receptor channels in the membrane by databases such as OPM suggest that much of the lateral fenestrations are buried in the membrane (*Figure 1*), the efficient reaction between MTS and E17C (*Figure 5*) suggests that this residue resides in an aqueous environment and is not buried in the membrane, supporting its location within the ion permeation pathway at the intracellular end of the pore. Importantly, charge-reversing mutations in rP2X2 and mP2X5 clearly alter the relative permeability of the two channels (*Figure 6* and *Figure 8*). Although the effect of the mutation in rP2X2 is unambiguous (*Figure 6*), the result also suggests that other determinants contribute to cation selectivity for this subtype (*Kawate et al., 2011*). In stark contrast, the influence of the mutations in mP2X5 (*Figure 8*) suggests that K17 is a major determinant of anion permeability in that subtype. Viewed collectively, our results support the idea that ion permeation occurs through the lateral fenestrations and that these structural elements play important roles in determining the ion selectivity of P2X receptor channels.

An important question remains about whether the cytoplasmic caps seen in structures of slowly desensitizing hP2X3$_{Slow}$ receptors in an ATP-bound open state (*Mansoor et al., 2016*) and those observed in both closed and ATP-bound open states of hP2X7 receptors (*McCarthy et al., 2019*) are present in other P2X receptor subtypes. Many residues at critical positions in the cytoplasmic cap are conserved in other P2X receptor subtypes (*Figure 2*; *Figure 2—figure supplement 1*), supporting the notion that the cap is a conserved feature of P2X receptor channels. Given that the structures of hP2X3$_{Slow}$ where the cap is seen contain three mutations corresponding to residues present in the slowly desensitizing P2X2 receptor, it would seem likely that the cap will also be present in P2X2 receptors. Our results with MTS accessibility and ion selectivity experiments in P2X2 are consistent with the structure of P2X3$_{Slow}$ containing the cytoplasmic cap (*Hausmann et al., 2014*; *Mansoor et al., 2016*), further supporting the presence of a cap in P2X2 receptors. Similarly, the critical influence of K17 in P2X5 on ion selectivity in that subtype would be consistent with the presence of cytoplasmic cap and lateral fenestrations like those present in the P2X3$_{Slow}$ structure. Although earlier studies had implicated residues in the outer pore as critical determinants of $Ca^{2+}$ permeability in P2X receptors (*Migita et al., 2001*; *Samways and Egan, 2007*; *Samways et al., 2014*), a recent study also implicated residues in the N-terminus of rP2X7 as determinants of $Ca^{2+}$ permeability (*Liang et al., 2019*). Indeed, the E17A mutation in rP2X2 and the equivalent E14A mutation in rP2X7 both diminish fractional $Ca^{2+}$ current (*Liang et al., 2019*), supporting a role of the lateral fenestrations in ion permeation and selectivity. In the future, it would be important to solve structures of other P2X receptor subtypes to ascertain whether the cytoplasmic cap and lateral fenestrations are features common to all subtypes, and to determine whether the cap is a transient structure, as proposed for P2X3 receptors (*Mansoor et al., 2016*) or a more stable one as proposed for P2X7 (*McCarthy et al., 2019*).

One of the unique features seen in structures of detergent-solubilized P2X receptor channels with ATP bound is that the lateral fenestrations present on both sides of the membrane, which enable ions to enter or exit the pore (*Kawate et al., 2011*), are continuous and present throughout the region likely spanning the lipid membrane (*Figure 1A and B*). As a result, the TM helices lack intersubunit interactions within the membrane and all intersubunit interactions occur within the large extracellular domain and within the cytoplasmic cap. Related membrane-embedded fenestrations have thus far not been observed in structures of other trimeric ion channels under conditions where the open state predominates, including acid-sensing ion channels (*Jasti et al., 2007*; *Gonzales et al., 2009*; *Baconguis et al., 2014*; *Yoder et al., 2018*; *Yoder and Gouaux, 2020*), epithelial sodium channels (*Noreng et al., 2018*; *Noreng et al., 2020*) or proton-activated chloride channels (*Ruan et al., 2020*; *Wang et al., 2022*). It will be interesting to solve structures of P2X receptor channels in a membrane environment to better understand whether membrane-embedded fenestrations are native features, whether lipids may play key structural roles in stabilizing membrane-spanning subunits or whether the fenestrations might distort the structure of the membrane to enable efficient permeation of ions through the internal end of the pore.

It will also be fascinating to explore how the ion selectivity of P2X receptor channel subtypes has been tuned to underlie distinct physiological functions in different cellular environments. Although, many important roles of cation-selective subtype of P2X receptors are emerging, the functions of the anion permeable P2X5 receptor remain largely enigmatic (*Khakh and North, 2006*; *Surprenant and North, 2009*; *Schmid and Evans, 2019*). P2X5 is thought to be widely expressed within neurons in the central nervous system (*Guo et al., 2008*) as well as in cardiac and skeletal muscle (*Garcia-Guzman*

*et al., 1996*; *Ryten et al., 2001*), yet what roles this channel plays will require further investigation. The unique ion selectivity of P2X5 receptors, determined by one critical residue within the lateral fenestrations, suggests that it may play very different roles than other P2X receptor subtypes.

## Materials and methods

### Channel constructs

Rat P2X2 (rP2X2) (*Brake et al., 1994*) cDNA in pcDNA1 was generously provided by Dr. David Julius (the University of California, San Francisco, CA). Mouse P2X5 (mP2X5) cDNA in pCMV6-Entry was purchased from OriGene and subcloned into pcDNA3.1. A previously characterized rP2X2 construct where three native cysteines (C9, C348, and C430) were mutated to threonine (rP2X2–3 T) was used as a background construct for MTS experiments because it is insensitive to MTS reagents yet displays functional properties that are similar to the wild-type channel (*Li et al., 2008*). All mutations in rP2X2 and mP2X5 were made using the QuikChange Lightning technique (Agilent Technologies) and confirmed by DNA sequencing (Macrogen).

### Cell culture

Human Embryonic Kidney (HEK293) cells were cultured in Dulbecco's modified Eagle's medium (DMEM) supplemented with 10% fetal bovine serum (FBS) and 10 mg L$^{-1}$ of gentamicin. HEK293 cells between passage numbers 5–20 were used and passaged when cells were between 40–80% confluent. The cells were treated with trypsin and then seeded on glass coverslips at about 15% of the original confluency in 35 mm Petri dishes. Transfections were done using the FuGENE6 Transfection Reagent (Promega). Transfected cells were incubated at 37 °C with 95% air and 5% $CO_2$ overnight for use in whole-cell recordings; 16–24 hr for reversal potential measurements and 40–48 hr for MTS experiments. All P2X constructs were co-transfected with a green fluorescent protein cDNA to identify transfected cells.

### Electrophysiology

Membrane currents were recorded from HEK293 cells using the whole-cell patch-clamp technique. When the whole-cell configuration was established, the desired extracellular solution was continuously perfused onto the cell through a gravity-fed perfusion system. Membrane voltage was controlled using an Axopatch 200B patch-clamp amplifier (Axon Instruments) and currents were digitized using a Digidata 1440 A interface board and pCLAMP 10 software. Membrane currents were collected with a sampling rate of 10 kHz and filtered at 2 kHz. The standard extracellular solution used to form GΩ seals contained 140 mM NaCl, 5.4 mM KCl, 2 mM $CaCl_2$, 0.5 mM $MgCl_2$, 10 mM HEPES, and 10 mM D-glucose, adjusted to pH 7.3 with 1 M NaOH, with an osmolality of about 300 mmol/kg. The standard pipette solution contained 140 mM NaCl, 10 mM HEPES, and 10 mM EGTA, adjusted to pH 7.0 with 1 M NaOH. The standard extracellular recording solution used for obtaining concentration-response relations and MTS accessibility experiments contained 140 mM NaCl and 10 mM HEPES, adjusted to pH 7.3 with 1 M NaOH, with an osmolality of about 275 mmol/kg. Reversal potential measurements were made using the standard extracellular solution and one containing 40 mM NaCl, 10 HEPES, and 175 mM D-glucose or sucrose, adjusted to pH 7.3 with 1 M NaOH, with an osmolality of about 270 mmol/kg. Bath and ground chambers were connected by an agar bridge containing 3 M KCl. Liquid junction potentials between internal and external solutions were measured and found to be within ±1.5 mV and all voltages were not corrected. Solutions containing ATP were freshly prepared for use on the same day. Stock solutions of 100 mM MTS reagents (bromide salt; Toronto Research Chemicals) were prepared daily in deionized water and stored on ice. MTS reagents were diluted to the desired concentration less than 2 min prior to the application of the reagent to individual cells.

### Data analysis

Concentration-response relationships for ATP were obtained for each mutant channel and the Hill equation fit to the data according to:

$$\frac{I}{I_{max}} = \frac{[ATP]^{n_H}}{[ATP]^{n_H} + EC_{50}^{n_H}}$$

where $I$ is the normalized current at a given ATP concentration, $I_{max}$ is the maximum normalized current, $EC_{50}$ is the concentration of ATP ([ATP]) that elicits half-maximal currents, and $n_H$ is the Hill coefficient.

Time constants for MTS modification ($\tau$) were obtained by fitting the relaxation of current inhibition by MTS reagents with a single exponential function (**Figure 5**) and modification rates ($R$) were calculated according to:

$$R = \frac{1}{\tau [M]}$$

where [M] is the concentration of the MTS reagent.

The relative permeability of $P_{Cl}/P_{Na}$ was estimated using the Goldman-Hodgkin-Katz equation (**Hille, 2001**):

$$V_{rev} = \frac{RT}{F} ln \left( \frac{P_{Na}[Na]_{out} + P_{Cl}[Cl]_{in}}{P_{Na}[Na]_{in} + P_{Cl}[Cl]_{out}} \right)$$

# Acknowledgements

We thank Angela Ballesteros, Surbhi Dhingra, and members of the Swartz laboratory for helpful discussions. This research was supported by the Intramural Research Program of the National Institute of Neurological Disorders and Stroke, NIH, Bethesda, MD to KJS (NS003018).

# Additional information

### Competing interests
Kenton J Swartz: Senior editor, eLife. The other authors declare that no competing interests exist.

### Funding

| Funder | Grant reference number | Author |
| --- | --- | --- |
| National Institute of Neurological Disorders and Stroke | NS003018 | Kenton J Swartz |

The funders had no role in study design, data collection and interpretation, or the decision to submit the work for publication.

### Author contributions
Stephanie W Tam, Conceptualization, Data curation, Formal analysis, Validation, Investigation, Visualization, Methodology, Writing – original draft, Writing – review and editing; Kate Huffer, Data curation, Formal analysis, Validation, Investigation, Visualization, Writing – original draft, Writing – review and editing; Mufeng Li, Conceptualization, Data curation, Formal analysis, Supervision, Validation, Investigation, Writing – original draft, Writing – review and editing; Kenton J Swartz, Conceptualization, Supervision, Funding acquisition, Visualization, Methodology, Writing – original draft, Project administration

### Author ORCIDs
Stephanie W Tam ⓘ http://orcid.org/0000-0001-5395-9408
Kate Huffer ⓘ http://orcid.org/0000-0001-5003-3140
Kenton J Swartz ⓘ http://orcid.org/0000-0003-3419-0765

### Decision letter and Author response
Decision letter https://doi.org/10.7554/eLife.84796.sa1
Author response https://doi.org/10.7554/eLife.84796.sa2

# Additional files

## Supplementary files
• MDAR checklist

## Data availability
All data needed to evaluate the conclusions in this paper are available in the main text and supplementary materials.

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
