## [Editor Report]

This study provides valuable insight into the molecular mechanism of ion selectivity in the broader family of ATP-gated P2X receptors. The experimental data are of high quality, the evidence supporting the conclusions is convincing, and the work will be of broad interest to biophysicists working on ion channel selectivity.

---

## [Decision Letter]

**Decision letter after peer review:**

Thank you for submitting your article "Ion permeation pathway within the internal pore of P2X receptor channels" for consideration by *eLife*. Your article has been reviewed by 3 peer reviewers, and the evaluation has been overseen by a Reviewing Editor and Richard Aldrich as the Senior Editor. The following individuals involved in the review of your submission have agreed to reveal their identity: Thomas Grutter (Reviewer #1); Zhaozhu Qiu (Reviewer #2); Alexandria N Miller (Reviewer #3).

Essential revisions:

The strengthen the conclusions of the work, the following essential revisions are requested:

1) To increase n (especially where n currently equals only 3) and include statistical analysis where appropriate.

2) Either demonstrate the reversibility of the MTS modifications or include a discussion on the lack thereof.

*Reviewer #1 (Recommendations for the authors):*

Although I am convinced by provided data, there are some issues that need to be addressed before acceptance.

1. I would recommend the authors increase the number of cells they provided in some Figures. For example, the authors provide n = 3 cells for E17C in Figure 3F. Although error bars are small, three cells are insufficient.

2. It seems that the same traces are displayed in Figures 3, 4, and 5 to show currents of E17C in the absence or presence of MTS compounds. I recommend showing another set of relevant traces (for example fitting traces in Figure 5).

3. Figure 2H and Figure 2 —figure supplementary 1D. It would be nice to add labels of residues E11 and E14 for hP2X3Slow and rP2X7, respectively.

*Reviewer #2 (Recommendations for the authors):*

1. Statistical analyses are necessary for comparing some groups in the data presented, for example, 3E, F, 4E, F, to support the conclusions. Some n numbers are relatively small (i.e. n = 3).

2. If the block by thiol-reactive reagents can be reversed by reducing agents, it will exclude other alternative explanations of the channel inhibition observed in the cysteine accessibility assay and further strengthen the conclusions. Sometimes, however, the inhibitory effects cannot be reversed by the treatment of reducing agents. In that case, the authors may discuss and comment on that.

*Reviewer #3 (Recommendations for the authors):*

1) In this study, the authors 1) use charged MTS reagents, 2) observe partial block with these reagents, and 3) observe changes to cation to anion permeability with charged residue mutations. I was wondering if the authors did an investigation of cation-to-anion permeability using MTS-modified rP2X2-3T. In particular, MTSET is a positively charged MTS reagent and, one would expect that the cation to anion permeability would shift upon MTSET modification of rP2X2. If the data is available, it would be worthwhile to include it in the manuscript.

2) An ion pore that is formed by a portion of a larger, membrane-embedded fenestration is unusual to me. Is there any evidence that this is shared by other channels in this structural subfamily, such as ASIC, eNaC, and/or PAC? This would be worth discussing.

3) In the introduction, the cytoplasmic ballast of P2X7 is mentioned but it was unclear to me that this is a structural feature likely not conserved among P2X channels. This needs to be clarified.

4) Please include this citation:

Jiang, L-H; Rassendren, F., Spelta, V., Surprenant, A. and North, R.A. Amino Acid Residues Involved in Gating Identified in the First Membrane-spanning Domain of the Rat P2X2 Receptor. Journal of Biological Chemistry, 2001.

---

## [Author Response]

Essential revisions:The strengthen the conclusions of the work, the following essential revisions are requested:1) To increase n (especially where n currently equals only 3) and include statistical analysis where appropriate.

This request concerns the MTS modification experiments presented in Figure 3 and 4. We collected data from many more cells where MTS reagents were applied, and inhibition measured in the presence or absence of ATP. We could have presented those results similarly to what appears in many related studies, however, we decided to adopt a more involved multiple pulse protocols illustrated in these figures to better document irreversible modification and clearly distinguish this from rundown of the ATP responses for a population of cells. The longer protocols, in particular for the closed state modification where we subsequently test for open state modification, resulted in only a fraction of cells surviving long enough to complete the entire protocol, and hence the relatively small n values. We have added additional data to the legend with a larger number of cells using a more conventional approach where we simply state the % inhibition values for MTS reagents and we have also added additional data for 4 cells where we examined open state modification at 0.5 mM MTSEA in Figure 4E. We have also added a statistical analysis to Figure 3 and Figure 4 and provided p-values in the legend.

2) Either demonstrate the reversibility of the MTS modifications or include a discussion on the lack thereof.

In our experience with MTS modification experiments (*1-4*) we consider the demonstration of irreversibility to be the most important feature for concluding that modification has occurred. We have observed reversible effects of MTS reagents due to pore blockade, for example, but at the concentrations used here we observed no effects for the background construct but irreversible inhibition for E17C, which we consider to be strong evidence for modification of that position by both MTSET and MTSEA. In earlier experiments identifying pore-lining residues we had attempted to reverse the actions of MTS reagents with reducing reagents but found that they made recordings unstable, and we have always been concerned by the fact that there are 5 disulfide bonds within the extracellular domain of P2X receptors that might complicate intepretations. It is also worth noting that this experiment speaks more to the accessibility of the reducing reagent than they do about whether modification has occurred. For these reasons, we have not included results with reducing reagents in our previous publications or in the present manuscript. We have added a statement to the Results section to alert the reader that we have not attempted to reduce the MTS aduct with E17 and to provide our reasoning.

1. M. Li, T. H. Chang, S. D. Silberberg, K. J. Swartz, Gating the pore of P2X receptor channels. *Nat Neurosci* 11, 883-887 (2008).

2. M. Li, T. Kawate, S. D. Silberberg, K. J. Swartz, Pore-opening mechanism in trimeric P2X receptor channels. *Nat Commun* 1, 44 (2010).

3. T. Kawate, J. L. Robertson, M. Li, S. D. Silberberg, K. J. Swartz, Ion access pathway to the transmembrane pore in P2X receptor channels. *J Gen Physiol* 137, 579-590 (2011).

4. G. Heymann *et al.*, Inter- and intrasubunit interactions between transmembrane helices in the open state of P2X receptor channels. *Proc Natl Acad Sci U S A* 110, E4045-4054 (2013).

Reviewer #1 (Recommendations for the authors):Although I am convinced by provided data, there are some issues that need to be addressed before acceptance.1. I would recommend the authors increase the number of cells they provided in some Figures. For example, the authors provide n = 3 cells for E17C in Figure 3F. Although error bars are small, three cells are insufficient.

See response to essential revision 1.

2. It seems that the same traces are displayed in Figures 3, 4, and 5 to show currents of E17C in the absence or presence of MTS compounds. I recommend showing another set of relevant traces (for example fitting traces in Figure 5).

We have added additional examples to Figure 3 and 4 that are unique compared to those shown in Figure 5.

3. Figure 2H and Figure 2 —figure supplementary 1D. It would be nice to add labels of residues E11 and E14 for hP2X3Slow and rP2X7, respectively.

We have added these residues and labeled them as requested.

Reviewer #2 (Recommendations for the authors):1. Statistical analyses are necessary for comparing some groups in the data presented, for example, 3E, F, 4E, F, to support the conclusions. Some n numbers are relatively small (i.e. n = 3).

We have added the requested statistics to panels E and F for Figure 3 and Figure 4. Also see response to essential revision 1.

2. If the block by thiol-reactive reagents can be reversed by reducing agents, it will exclude other alternative explanations of the channel inhibition observed in the cysteine accessibility assay and further strengthen the conclusions. Sometimes, however, the inhibitory effects cannot be reversed by the treatment of reducing agents. In that case, the authors may discuss and comment on that.

See response to essential revision 2. As described above, we are confident about concluding that E17C is covalently modified by MTS reagents as the effects are irreversible and only seen in when the Cys has been introduced. With these results in hand, failure of reducing agents to reverse the effects of MTS reagents would speak more to accessibility of the reducing reagents rather than whether the MTS reagent had modified the introduced Cys.

Reviewer #3 (Recommendations for the authors):1) In this study, the authors 1) use charged MTS reagents, 2) observe partial block with these reagents, and 3) observe changes to cation to anion permeability with charged residue mutations. I was wondering if the authors did an investigation of cation-to-anion permeability using MTS-modified rP2X2-3T. In particular, MTSET is a positively charged MTS reagent and, one would expect that the cation to anion permeability would shift upon MTSET modification of rP2X2. If the data is available, it would be worthwhile to include it in the manuscript.

This is a very cool experiment and one that could be done with both positively and negatively charged MTS reagents. Unfortunately, we ran out of time to conduct these experiments before Stephanie Tam and Mufeng Li left the lab.

2) An ion pore that is formed by a portion of a larger, membrane-embedded fenestration is unusual to me. Is there any evidence that this is shared by other channels in this structural subfamily, such as ASIC, eNaC, and/or PAC? This would be worth discussing.

This is a unique feature of P2X receptor channels. Our original view was that the fenestrations at the edges of the membrane are likely to be native features of these channels, but that the separation of subunits within the membrane might be non-native and the result of detergent solubilization. It will be interesting to see whether these features are retained when structures are solved in lipid nanodiscs. As suggested, we have added a section to the discussion to highlight this intriguing feature of the available P2X receptor structures.

3) In the introduction, the cytoplasmic ballast of P2X7 is mentioned but it was unclear to me that this is a structural feature likely not conserved among P2X channels. This needs to be clarified.

Yes, the ballast is a feature unique to P2X7 receptor channels and we now make this clear in the introduction.

4) Please include this citation:Jiang, L-H; Rassendren, F., Spelta, V., Surprenant, A. and North, R.A. Amino Acid Residues Involved in Gating Identified in the First Membrane-spanning Domain of the Rat P2X2 Receptor. Journal of Biological Chemistry, 2001.

We have added the requested citation.